# Screening for Perinatal Depression: Barriers, Guidelines, and Measurement Scales

**DOI:** 10.3390/jcm13216511

**Published:** 2024-10-30

**Authors:** Kathleen A. Kendall-Tackett

**Affiliations:** Texas Tech University Health Sciences Center, Amarillo, TX 79106, USA; kkendallt@gmail.com

**Keywords:** pregnancy, perinatal, postpartum, postnatal, depression, screening, Edinburgh Postnatal Depression Scale, Patient Health Questionnaire

## Abstract

**Background:** Screening for perinatal depression can lower its prevalence and ensure that mothers receive adequate treatment and support. Yet, few practitioners screen for it. The present article is a brief review of barriers to screening, and two screening scales are validated for perinatal women. **Findings:** Even though health organizations recommend screening, most new mothers are not screened. Providers cite a lack of time, opening “Pandora’s box,” and a lack of resources for mothers who screen positive as the reasons why they do not screen for this condition. The Edinburgh Postnatal Depression Scale and the Patient Health Questionnaires are brief screening scales validated for new mothers and widely available. **Conclusions:** Screening is necessary to identify depression in pregnant and postpartum women. Practitioners who screen for this condition need a clear plan and knowledge of how to access available community resources so that they know what to do when a mother screens positive.

## 1. Introduction

Screening is necessary for identifying and treating mothers affected by perinatal depression. Hahn-Holbrook et al. (2018)’s [1] systematic review included 291 studies from 56 countries (*N* = 296,284). The worldwide rate for postpartum depression was 18%. A recent US population study, collecting data from 31 sites (Bauman et al., 2020) [2], found that 13% of mothers were depressed at 3 and 4 months postpartum. However, this aggregate number seriously underestimates the depression rates in some groups. For example, 11% of White women were depressed, but the depression rates for women of color ranged from 18% to 22%.

Screening for depression during pregnancy is important and not often carried out. Yet, during pregnancy, women are routinely screened for gestational diabetes and pregnancy-induced hypertension, which affect far fewer mothers than depression, which, among other things, is also a known risk factor for preterm birth (Grigoriadis et al., 2018; Nutor et al., 2018) [3,4].

Screening is also important during the postpartum period. The US Agency for Healthcare Research and Quality (AHRQ) found that screening increases the likelihood that postpartum depression will be treated or even prevented (Myers et al., 2013) [5]. A review of 23 studies (*N* = 5398) found that screening leads to lower rates of postpartum depression and increased treatment or remission, especially if screening is followed by clinical care (O’Connor et al., 2016) [6]. Screening scales particularly help when mothers are reluctant to tell others that they are depressed or when they believe their symptoms to be a “normal” part of new motherhood (Sword et al., 2008) [7]. In one study, only 34% of depressed mothers sought help for their condition (McGarry et al., 2009) [8].

Unfortunately, untreated depression can lead to serious consequences. During pregnancy, depression increases the risk of negative birth outcomes. After delivery, untreated depression can affect mothers’ interactions with their infants, their relationships with their partners, and their sleep (Kendall-Tackett et al., 2011) [9]. The two most serious potential outcomes are suicide and infanticide (Kendall-Tackett, 2023) [10].

Yet, despite these warnings and the recommendations of major health organizations, most pregnant and postpartum women are not screened. This article briefly reviews recent studies on barriers to screening for perinatal depression, ways to incorporate screening into clinical practice, and information about the two most widely used screening tools: the Edinburgh Postnatal Depression Scale and the Patient Health Questionnaires.

## 2. Screening Guidelines

Health organizations that treat perinatal women acknowledge that screening is important, but some professional guidelines could be more specific and helpful. For example, the American College of Obstetricians and Gynecologists (2023) [11] recommends that “routine screening by physicians is important for ensuring appropriate follow-up and treatment” but does not provide any details about what providers must do. Busy practitioners are not likely to screen when given such generic advice.

A better example of screening guidelines comes from the UK’s National Institute for Health and Clinical Excellence (NICE). NICE tells providers who should perform the screen (the primary provider), when screening should occur (upon first contact, at 4 to 6 weeks and 3 to 4 months), which providers should screen (all), and which questions to ask (National Institute for Health and Care Excellence (NICE), 2020) [12].

## 3. Barriers to Screening

Healthcare providers cite many reasons for not screening, including a lack of resources and opening a “Pandora’s box” (i.e., questions might reveal a situation which providers are unequipped to handle, so it is better not to ask). Canada′s national health system *recommends that providers not screen* because policymakers worry that false positives could swamp their providers (Lang et al., 2022) [13]. This recommendation is troubling. The authors provide no statistics on the number of false positives and whether they are a problem. In addition, nearly a quarter of new Canadian mothers have either postpartum depression or anxiety (Statistics Canada, 2019) [14]. How will these mothers be identified and treated?

The lack of screening can also be differentially harmful to racial/ethnic minority women. In US samples, racial/ethnic minority women are even less likely to be screened for depression. When they are, their rates of depression, anxiety, and PTSD are much higher than among White women. The Listening to Mothers in California study included a representative sample of 2539 women (Declercq et al., 2021) [15]. Black mothers had higher rates of depression and were less likely to be treated (via medication or counseling) compared to White women. However, when women were screened, they were six times more likely to receive counseling. The authors recommended universal screening to reduce health inequities and increase the rates of treatment, especially for non-White women.

## 4. Screening Scales

This review focuses on the Edinburgh Postnatal Depression Scale (EPDS) and the Patient Health Questionnaires (PHQs). Other scales that appear in studies of perinatal women include the following two diagnostic scales: the Hamilton Rating Scale for Depression (HAM-D) and the Center for Epidemiologic Studies-Depression (CES-D). The EPDS and PHQs have both been validated against these scales. The Beck Depression Inventory is less commonly used and not specific to pregnancy or postpartum.

The American Academy of Pediatrics, the US Preventive Services Task Force, and a review in the *American Family Physician* recommend both the EPDS and PHQs (Earls and Committee on Psychosocial Aspects of Child and Family Health, 2010 [16]; Maurer et al., 2018 [17]; Rafferty et al., 2019 [18]; Siu and US Preventive Services Task Force (USPSTF), 2016 [19]). However, both measures are *screening scales* and should not be used to diagnose depression, although they are frequently used to diagnose it. The EPDS and PHQs screen for *symptoms* of depression or anxiety but do not identify risk factors. Risk factors should be identified in follow-up conversations, often with licensed mental healthcare providers. Both scales have been validated with perinatal samples and have advantages and limitations. These scales and all their variations are available for free online.

The symptoms of perinatal depression are the same as those for major depressive disorder in the *Diagnostic and Statistical Manual−5 Text Revision* (American Psychiatric Association, 2013, 2022) [20,21]. The most common symptoms are depressed mood and anhedonia. For a diagnosis of major depression, women must have at least five symptoms, and they must last for at least 2 weeks. In addition, perinatal depression often co-occurs with anxiety. A study of new mothers from Croatia (*N* = 272) found that 75% of depressed new mothers also had anxiety (Rados et al., 2018) [22].

### 4.1. The Edinburgh Postnatal Depression Scale (EPDS)

The Edinburgh Postnatal Depression Scale (EPDS) is the most widely used postpartum depression screening tool in the world (Cox et al., 1987) [23]. It is a 10-item self-report questionnaire. The EPDS has been validated worldwide and has been translated into many languages. The EPDS asks women to report how they have felt *in the past week*, and each answer is scored from 0 to 3. The EPDS was developed to address postpartum women specifically, unlike previous scales.

#### 4.1.1. Validation of the EPDS

To validate a new scale, researchers compare its results to those of established scales that measure the same construct (e.g., depression). When the new scale identifies approximately the same people as having depression as an established scale, researchers consider it a valid measure. The EPDS has been validated against the Center for Epidemiologic Studies-Depression Scale (CES-D). The CES-D is more detailed and can be used for diagnosis, but the results are consistent with the EPDS (Logsdon et al., 2009) [24]. When used 6 weeks postpartum, the EPDS performs as well as the Hamilton Rating Scale for Depression-17 and -21 and the Beck Depression Inventory (Myers et al., 2013) [5].

Another study used the Danish version of the EPDS and compared it to DSM-5 and ICD-10 diagnostic criteria for postpartum depression (Smith-Nielsen et al., 2018) [25]. They had 324 women in their sample and assessed them at 2 and 10 months postpartum. Eleven was the most accurate cutoff to validate the EPDS against the DSM-5 and International Classification of Diseases, 10th Edition (ICD-10) criteria (World Health Organization, 2019) [26].

The EPDS is also for non-White, non-middle-class samples. In a sample of 169 low-income African American women, King (2012) found that many of them were depressed. This sample is important because Black women are frequently underrepresented in postpartum depression studies [27]. This study also demonstrated the differences between high vs. low cutoffs. With a cutoff of ≥10, 30% were depressed. When the cutoff was higher (>13, i.e., more severe depression), 19% were depressed. In addition, her results indicated that the EPDS measures more than depression. A confirmatory factor analysis demonstrated that the EPDS measured depression, anxiety, and anhedonia. Shorter versions, such as the EPDS-3, focus specifically on anxiety symptoms (Kabir et al., 2008) [28]. These studies are described in the next section.

#### 4.1.2. Scale Cutoffs

Twelve is the standard EDPS cutoff (Cox, 2019) [29], but, as King (2012)’s study demonstrated, researchers often use higher and lower cutoffs depending on what they want to achieve [27]. The cutoff can vary depending on its intended purpose: for instance, broad vs. narrow screening. Broad screening, with lower cutoffs (EPDS score = 9–11), identifies women with milder symptoms and some who are not depressed (i.e., false positives). Narrow screening, with higher cutoffs (EPDS score = 12–14+), identifies women with major depression but misses women with milder symptoms. Which cutoff you use depends on your goals. If you wanted to identify all depressed women in your population, a lower cutoff would be better. For example, Dennis (2004) [30] found that an EPDS cutoff of 9 was more sensitive, making it a better choice for community samples. In contrast, higher cutoff scores increase specificity (i.e., only identify women with severe symptoms) but miss depressed women with milder symptoms. For example, when Dennis et al. (2004) [31] used a cutoff of 12/13 in her study of 594 women, she failed to identify 43% of mothers who were depressed at 4 weeks and 53% of mothers who were depressed at 8 weeks.

Lewis et al. (2020)’s review of 58 studies on new mothers (*N* = 15,557) found that 2069 of them had severe depressive symptoms. A cutoff of 11 led to the highest sensitivity and specificity [32]. Lewis et al. found the same pattern across countries, but most studies were not from low- or middle-income countries [32].

Dennis et al. (2016) [33], using a stratified sample of 1125 new Canadian mothers, compared cutoffs of 9 vs. 12 at 1 and 16 weeks postpartum. A cutoff of 9 was more sensitive and predicted which mothers were likely to be depressed at 16 weeks. There were 64 false positives, but no one was harmed by being misidentified as being depressed at 1 week. Higher cutoff scores (12/13) decrease the false positives but increase the false negatives.

Conversely, a practitioner or program with limited resources, who can only treat women with more severe symptoms, may find a higher cutoff to be more appropriate. A cutoff of 13 or more had a sensitivity of 0.8 and a specificity of 0.9 in a review of 23 studies (*N* = 5398) that used the EPDS in English (O’Connor et al., 2016) [6]. A recent meta-analysis compared the EPDS to the diagnostic Structured Clinical Interview for DSM (SCID) (Lyubenova et al., 2021) [34]. The combined sample from 29 studies (*N* = 7315 women) found that the SCID identified 1017 women (9%) with major depression. The studies compared those scores to the EPDS. Lower cutoffs identified more women as being depressed compared to higher cutoffs. Fourteen, the most stringent cutoff on the EPDS, was the most accurate compared to the SCID scores. However, the SCID specifically identifies only severe depression. When trying to also identify mild and moderate depression with lower cutoffs, the EPDS was not as accurate compared to the SCID. Still, it did correctly identify women with milder symptoms.

#### 4.1.3. Advantages and Disadvantages of the EPDS

The advantages of the EPDS include the fact that it is easy to complete, that mothers can complete the scale in a few minutes, and that it is acceptable to use it in research because it makes comparing results across published articles easier. The disadvantages include awkward and colloquial wording and the potential for scoring mistakes.

#### 4.1.4. Language Issues

The EPDS can be difficult to translate accurately because the items are in colloquial British English. When translated word for word, they can be challenging for mothers outside the UK or the Commonwealth to understand, ultimately leading to false negatives (Waqas et al., 2021) [35]. This issue was highlighted in a review of 16 studies with mothers in low-to-middle-income countries (LMIC; *N* = 1281) (Shrestha et al., 2016) [36]. In most cases, versions in the local language were less accurate than English ones. Only 1 study out of 16 used a culturally sensitive translation that employed local expressions to describe depression and anxiety. Culturally sensitive versions were twice as accurate.

Literal translations are also more difficult for mothers who have low literacy levels. Shrestha et al. (2016) [36] recommended that cutoff scores be validated for each culture along with a culturally validated standard diagnostic protocol. An inaccurate measure may mean that mothers with depression are not identified.

“Translation” issues for the EPDS even happen for American mothers (Cox, 2019) [29]. American mothers often find the wording of some questions odd (e.g., “Things are getting on top of me lately”). Unfortunately, a colloquial British expression that does not consider how American mothers describe being overwhelmed would not accurately assess their feelings. A new US version of the EPDS is now available, which may increase accuracy for American mothers (Moyer et al., 2023) [37].

#### 4.1.5. Scoring Issues

Scoring is another challenge. Several items on the EPDS use reverse scoring, increasing instances of mis-scoring. Matthey et al. (2013) [38] examined 496 EPDSs from six Australian practices. Twenty-two clinicians estimated their rate of errors and learned that they had made many more scoring mistakes than they had originally estimated: 17% had at least one scoring error. Most scores were only one point off, meaning a false negative or positive.

#### 4.1.6. The Edinburgh Postnatal Depression Scale-3 (EPDS-3)

Several researchers have used shorter versions of the EPDS and have found that these are often more accurate, particularly for anxiety symptoms (Walker et al., 2015) [39]. One study used three items that assessed anxiety and compared the three-item version to the full-scale in a sample of 199 new mothers, aged 14–26 (Kabir et al., 2008) [28]. With a cutoff of 10, 21% met the criteria for depression. The EPDS-3 had a 95% sensitivity and identified 16% more depressed mothers as depressed compared to the full scale. The questions on the 3-item version included:I have blamed myself unnecessarily when things went wrong. (item 3)I have felt scared and panicky for no very good reason. (item 5)I have been anxious or worried for no good reason. (item 4)

Kabir et al. suggested incorporating the EPDS-3 into well-baby checks [28]. Similarly, King (2012) [27] found that the anxiety questions best predicted postpartum depression at 4 to 6 weeks in her sample of African American mothers. The anxiety items included items 3, 4, and 5, and two additional items: inability to cope (item 6) and difficulty sleeping (item 7).

#### 4.1.7. Summary

The EPDS is widely available, reliable, and valid. There are scoring and language challenges with the EPDS, but these can be overcome. Even with these limitations, the EPDS is an accurate measure of depression in new mothers and is most sensitive with a lower cutoff score.

### 4.2. The Patient Health Questionnaires

Many health organizations have widely adopted the Patient Health Questionnaires (PHQs) in their two-, four-, and nine-item versions (Sun et al., 2020) [40]. The Patient Health Questionnaires (PHQs) are available for free. They are written in American English, and the items are phrased in less colloquial English, making them easier to understand and translate. These scales were developed for the general population and validated with pregnant and postpartum women.

#### 4.2.1. Patient Health Questionnaire-9 (PHQ-9)

The PHQ-9 is the master scale and includes questions about the nine symptoms in the diagnostic criteria for major depression (American Psychiatric Association, 2013, 2022) [20,21]. It has high specificity and sensitivity. The samples used to validate the scale included 3000 patients from primary care and 3000 from OB/Gyn, with high measures of internal validity (Cronbach’s alpha) and test–retest scores for both samples (Kroenke et al., 2001) [41]. The PHQ-9 was also compared to the depression questions from the Pregnancy Risk Assessment Monitoring Systems (PRAMS-6) (Davis et al., 2013) [42], which were validated against the Structured Clinical Interview for Depression (SCID) and the Hamilton Rating Scale for Depression (HAM-D). It was less accurate than the SCID but more accurate than the PRAMS-6.

#### 4.2.2. Patient Health Questionnaire-2 (PHQ-2)

The two-item Patient Health Questionnaire (PHQ-2) contains two items from the PHQ-9, has been validated for perinatal women, and is recommended by the American Academy of Pediatrics (AAP) and the American College of Obstetricians and Gynecologists (ACOG) (Earls and Committee on Psychosocial Aspects of Child and Family Health, 2010 [16]; Walker et al., 2015 [39]). If, however, mothers score higher than 3 (possible depression), the AAP and ACOG recommend using the full PHQ-9.

The PHQ-2 items ask about how often mothers have experienced anhedonia (“little interest or pleasure in doing things”) and depressed mood (“feeling down, depressed or hopeless”) during *the past two weeks*. The response categories include Not at All, Several Days, More than Half the Days, and Nearly Every Day. The US Centers for Disease Control used the PHQ-2 in its most recent Pregnancy Risk Assessment Monitoring Systems (PRAMS) to assess depression at 3 to 4 months postpartum (Bauman et al., 2020) [2].

However, a study from Kenya including 3605 women found that the PHQ-2 was less accurate than the Center for Epidemiologic Studies Depression Scale (CESD-10), the Edinburgh Postnatal Depression Scale (EPDS), and the PHQ-9 (Larsen et al., 2023) [43]. The PHQ-2 was the least accurate in detecting moderate to severe depression compared to the other scales.

A recent Australian study compared the EPDS and PHQ-2 with a sample of 252 pregnant women (Slavin et al., 2020) [44]. The PHQ-2 did not detect women who were depressed but had a high specificity with cutoffs of 2 or 3. After assessing women four times during pregnancy and postpartum, the authors concluded that the PHQ-2 missed too many women with major depression. One limitation that might account for this inaccuracy is that anxiety is not assessed, a limitation which is addressed by the PHQ-4.

#### 4.2.3. Patient Health Questionnaire-4 (PHQ-4)

The PHQ-4 contains the items from PHQ-2 and adds two questions from the Generalized Anxiety Disorder-2 scale (GAD-2) (Kroenke et al., 2009) [45]. Scoring 3 or more on the first two questions suggests possible anxiety. A score of 3 or more on questions 3 and 4 indicates possible depression (Wicke et al., 2022) [46]. The PHQ-4 accurately predicts anxiety disorders but not post-traumatic stress disorder. A recent study of 6874 women from 64 countries used the PHQ-4 to understand the impact of giving birth during COVID-19 (Basu et al., 2021) [47]. Thirty-one percent were above the cutoff for depression or anxiety.

The Listening to Mothers in California survey (*N* = 2539) also used the PHQ-4 and found that it accurately assessed all mothers, including racial/ethnic minorities (Declercq et al., 2021) [15]. Another perinatal study used the PHQ-4 in an 11-year longitudinal study of 20 women with osteoporosis related to pregnancy and breastfeeding (Gehlen et al., 2019) [48].

A Spanish study included 845 pregnant women recruited from public hospitals (Rodriguez-Munoz et al., 2020) [49]. The authors examined the validity and psychometric properties of the PHQ-4 for pregnant women. Using exploratory and confirmatory factor analysis, the PHQ-4 accurately assessed depression and anxiety. They concluded that the PHQ-5 was a valid measure for pregnant women.

Similarly, a Pakistani study with mothers from rural areas compared three screening scales: the PHQ-4, the four-item Hamilton Depression Rating Scale, and the Community Informant Detection Tool (Waqas et al., 2021) [35]. Each measure worked, but the PHQ-4 was particularly accurate, with a sensitivity of 93% and a specificity of 92%. An American study also used the PHQ-4 in a sample that included 1148 pregnant women who spoke either English or Spanish (Barrera et al., 2021) [50]. A confirmatory factor analysis demonstrated that the PHQ-4 was reliable and valid for pregnant women.

#### 4.2.4. Summary

Like the EPDS, the PHQs have solid track records as assessment tools for new mothers. The PHQ-9 and PHQ-4 have better accuracy. The PHQ-2 can be used if its limitations are recognized and addressed.

## 5. Conclusions

Screening must take place before treatment can occur. Mothers cannot be treated if practitioners do not know they are depressed. Health organizations recognize the importance of screening and recommend it. Unfortunately, most new mothers are not screened. The guidelines are often vague, and practitioners lack practical advice about what to do when mothers screen positive. Barriers to screening include a lack of time, no resources to refer mothers to, and a fear of “Pandora’s box.” When administrators mandate screening but do not follow up to ensure that it is performed, practitioners are unlikely to carry it out. To increase the screening rate, organizations must ensure that providers have “buy-in” and understand why it is important to identify these mothers before their symptoms become more severe. Electronic prompts in medical records also increase screening in health settings.

Many practitioners and health organizations use the EPDS because it is so common and widely available, but this may not be appropriate for mothers in their communities. Providers should also be aware of possible scoring mistakes and how to safeguard against them. Lower cutoffs identify more depressed mothers, so communities with scarce resources might opt for a higher cutoff, allowing them to only triage mothers with more severe symptoms. The EPDS-3 can be a good alternative and is easier to complete and score than the full scale.

The Patient Health Questionnaires are excellent assessment tools for screening perinatal women. Many practitioners prefer the PHQs over the EPDS because the language is less colloquial, and they are easier to score. The PHQ-4 and -9 are more accurate than the PHQ-2.

Another issue is whether the EPDS and PHQs accurately measure depression in racial/ethnic minority populations. Heck (2018)’s review of 59 studies noted that both scales missed depression in racial/ethnic groups [51]. However, a US study using the PHQ-2—the weakest scale—found higher rates of depression in all racial/ethnic minority groups compared to White women (Bauman et al., 2020) [2]. While there is always room for improvement, these scales may be more accurate in assessing ethnic minority mothers than researchers have previously believed.

### Screening Is the First Step—What Happens Next?

After screening, providers need to act. According to the US Preventive Services Task Force, screening for depression should take place in a context allowing for accurate diagnosis, effective treatment, and appropriate follow-up (O’Connor et al., 2016) [6]. Yeaton-Massey and Herrero (2019) [52] recommend education, support, and timely resource referral. From a practical perspective, this means knowing what to do if a mother screens positive. Unfortunately, some practitioners engage in magical thinking about depression. They believe that including a screening scale in their assessments “fixes” depression. It does not. Screening without follow-up is useless.

If you are considering a screening program with your patients, your first step should be identifying community resources that can help. Plenty of evidence-based treatments for depression, anxiety, and PTSD are available (Kendall-Tackett, 2024) [53]. Be sure to include online resources. Mothers can be referred to follow-up medical care as needed.

The goal of screening is early detection and the treatment of symptoms. Catching depression early can stop its escalation into more severe symptoms and can save mothers and their families from years of misery. It is very much worth the effort.

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
