# Peer review of "Screening for Perinatal Depression: Barriers, Guidelines, and Measurement Scales"

_jcm, 2024, doi:10.3390/jcm13216511_

Round 1

Reviewer 1 Report

Comments and Suggestions for Authors

I consider this manuscript very interesting and accurate. In my opinion it only needs some minor revisions, first I would ask the authors to stress and strengthen the concept of social and continental differences in postpartum depression and secondly I would ask to mention the clinical correlation that has recently been studied between pCRH and the risk of postpartum depression (Almeida IF, Rinne GR, Coussons-Read M, Dunkel Schetter C. Placental corticotrophin-releasing hormone trajectories in pregnancy: Associations with postpartum depressive symptoms. Psychoneuroendocrinology. 2024 Jun;164:107030. doi: 10.1016/j.psyneuen.2024.107030. Epub 2024 Mar 21. PMID: 38537413.) and also the study of genetic polymorphisms and the risk of postpartum depression (Gutiérrez-Zotes A, Díaz-Peña R, Costas J, Martorell L, Gelabert E, Sans T, Navinés R, Albacar G, Ímaz ML, García-Esteve L, Sanjuan J, Martín-Santos R, Carracedo A, Vilella E. Interaction between the functional SNP rs2070951 in NR3C2 gene and high levels of plasma corticotropin-releasing hormone associates with postpartum depression . Arch Womens Ment Health. 2020 Jun;23(3):413-420. doi: 10.1007/s00737-019-00989-x. Epub 2019 Aug 6. PMID: 31388769.)

Author Response

I consider this manuscript very interesting and accurate. In my opinion it only needs some minor revisions, first I would ask the authors to stress and strengthen the concept of social and continental differences in postpartum depression and secondly I would ask to mention the clinical correlation that has recently been studied between pCRH and the risk of postpartum depression (Almeida IF, Rinne GR, Coussons-Read M, Dunkel Schetter C. Placental corticotrophin-releasing hormone trajectories in pregnancy: Associations with postpartum depressive symptoms. Psychoneuroendocrinology. 2024 Jun;164:107030. doi: 10.1016/j.psyneuen.2024.107030. Epub 2024 Mar 21. PMID: 38537413.) and also the study of genetic polymorphisms and the risk of postpartum depression (Gutiérrez-Zotes A, Díaz-Peña R, Costas J, Martorell L, Gelabert E, Sans T, Navinés R, Albacar G, Ímaz ML, García-Esteve L, Sanjuan J, Martín-Santos R, Carracedo A, Vilella E. Interaction between the functional SNP rs2070951 in NR3C2 gene and high levels of plasma corticotropin-releasing hormone associates with postpartum depression . Arch Womens Ment Health. 2020 Jun;23(3):413-420. doi: 10.1007/s00737-019-00989-x. Epub 2019 Aug 6. PMID: 31388769.)

This suggested articles are very interesting, and I highlight them in my most recent book, Depression in New Mothers, 4th Ed, Vol I: Causes, consequences, and risk factors. Unfortunately, it would not make sense to add them to the current paper because it focuses on screening for depression, with a review of various tools. The suggested references describe risk factors, which these scales do not measure. These scales only measure symptoms of depression. However, I do find the inflammatory aspect of PPD to be highly relevant and have written extensively about it--it is just not for this article. I will add a sentence to the article mentioning that these scales do no identify risk factors, but only symptoms.

Reviewer 2 Report

Comments and Suggestions for Authors

Thank you for the opportunity to read this study, which explores postpartum depression, focusing on screening barriers and tools. Despite the clinical importance of the work, other recent manuscripts analyse the topic (e.g., Ukatu et al., 2018, Psychosomatics;  Simas et al., 2023, JAMA). Below, I report some comments/suggestions on whether it could be considered for publication (maybe as a brief clinical review).

General comment: I would like to read at the beginning of the manuscript a definition of postpartum depression, similar/different symptoms compared to major depression, prevalence data, and the reasons of the importance of the screening (e,g,, prevent psychological problems, improving the quality of life of newborns, etc.).

Lines 31-32: I would explore other used tools in literature and clinical practice, explaining why it will be or not be appropriate to measure postpartum depressive symptoms.

Lines 35-47: Are there guidelines that suggest the assessment of depressive symptoms also during pregnancy? Many studies recently indicated that the most important predictor of postpartum depression is the presence of depressive symptoms (or other psychological disorders) during pregnancy.

Lines 50-67: I would enhance this section a bit, focusing on other barriers to screening (e.g., stigma, new mothers not wanting to be screened, time issues, available services and professionals to screening, etc.).

Lines 331-345: I would like more comments on this section. It is very important from a clinical perspective and an excellent idea to report the next steps after the screening. The author could better describe the most appropriate steps after the screening (e.g., semi-structured/structured interview, sending the patient to a mental health expert, proposing support groups, etc.).

Author Response

Thank you for the opportunity to read this study, which explores postpartum depression, focusing on screening barriers and tools. Despite the clinical importance of the work, other recent manuscripts analyse the topic (e.g., Ukatu et al., 2018, Psychosomatics;  Simas et al., 2023, JAMA). Below, I report some comments/suggestions on whether it could be considered for publication (maybe as a brief clinical review).

General comment: I would like to read at the beginning of the manuscript a definition of postpartum depression, similar/different symptoms compared to major depression, prevalence data, and the reasons of the importance of the screening (e,g,, prevent psychological problems, improving the quality of life of newborns, etc.).

I will add a brief summary of the symptoms, prevalence, and problems associated with depression. 

Lines 31-32: I would explore other used tools in literature and clinical practice, explaining why it will be or not be appropriate to measure postpartum depressive symptoms.

I can do this briefly but I need to be mindful of the word count limit. Honestly, the EPDS and PHQ are the two most often recommended scales. But I will briefly mention a few others.

Lines 35-47: Are there guidelines that suggest the assessment of depressive symptoms also during pregnancy? Many studies recently indicated that the most important predictor of postpartum depression is the presence of depressive symptoms (or other psychological disorders) during pregnancy.

Yes, these scales can also be used during pregnancy and the PHQs have been specifically validated for use with pregnant women. And depression in pregnancy is an important risk factor, but partner violence is now the highest risk factor, having surpassed depression in pregnancy. Both are important but it is no longer the highest.

Lines 50-67: I would enhance this section a bit, focusing on other barriers to screening (e.g., stigma, new mothers not wanting to be screened, time issues, available services and professionals to screening, etc.).

I can add more here, but the biggest barrier is providers not being willing to do it and/or offering no follow-up. Most studies of patients' attitudes find that they are generally positive about being screened. Whether they answer honestly is another issue ... 

Lines 331-345: I would like more comments on this section. It is very important from a clinical perspective and an excellent idea to report the next steps after the screening. The author could better describe the most appropriate steps after the screening (e.g., semi-structured/structured interview, sending the patient to a mental health expert, proposing support groups, etc.).

I will increase this section, but a lot depends on who is doing the screening and what resources are available. If it is at the community level, they may not have easy access to mental health care. If it is done by medical providers, they have access to other services but may not know what those are and have never researched it. I will add this to the discussion.

Reviewer 3 Report

Comments and Suggestions for Authors

I  am grateful for the chance to review this interesting article. The article presents a review of the literature on postpartum depression. The article provides a comprehensive examination of the obstacles, challenges, and recommendations for the assessment and diagnosis of postpartum depression. A comprehensive examination is conducted of the primary challenges, the discrepancies in detection scales, and recommendations for enhancing the detection of postpartum depression and its clinical significance. The conclusions are consistent with the information presented. In general, the article provides a comprehensive and valuable review of the available information on the subject. As a suggestion for further consideration, it may be beneficial to discuss and analyze the role played by gender bias in mental health care, as well as the implicit bias in diagnostic tools and in the general care of health providers.

Author Response

I  am grateful for the chance to review this interesting article. The article presents a review of the literature on postpartum depression. The article provides a comprehensive examination of the obstacles, challenges, and recommendations for the assessment and diagnosis of postpartum depression. A comprehensive examination is conducted of the primary challenges, the discrepancies in detection scales, and recommendations for enhancing the detection of postpartum depression and its clinical significance. The conclusions are consistent with the information presented. In general, the article provides a comprehensive and valuable review of the available information on the subject.

Thank you for your kind comments. I appreciate it.

As a suggestion for further consideration, it may be beneficial to discuss and analyze the role played by gender bias in mental health care, as well as the implicit bias in diagnostic tools and in the general care of health providers.

For gender bias, I would mostly be guessing since there really hasn't been much on it (surprisingly). There have been some studies on implicit bias using the scales with different groups. I can address that, particularly with the EPDS.